# JOINTLY LEARNING TOPIC SPECIFIC WORD AND DOCUMENT EMBEDDING

## ABSTRACT

Document embedding generally ignores underlying topics, which fails to capture polysemous terms that can mislead to improper thematic representation. Moreover, embedding a new document during the test process needs a complex and expensive inference method. Some models first learn word embeddings and later learn underlying topics using a clustering algorithm for document representation; those methods miss the mutual interaction between the two paradigms. To this point, we propose a novel document-embedding method by weighted averaging of jointly learning topic-specific word embeddings called TDE: Topical Document Embedding, which efficiently captures syntactic and semantic properties by utilizing three levels of knowledge -i.e., word, topic, and document. TDE obtains document vectors on the fly simultaneously during the jointly learning process of the topical word embeddings. Experiments demonstrate better topical word embeddings using document vector as a global context and better document classification results on the obtained document embeddings by the proposed method over the recent related models.

## 1 INTRODUCTION

In recent years the neural network-based methods have shown great success in various NLP-related tasks (Zhao et al., 2021). Tow popular complementary natural language processing methods are word2vec (Mikolov et al., 2013a;b) and Latent Dirichlet Allocation (LDA)(Blei et al., 2003); but executing a model in two-step lacks mutual interaction between the two paradigms and fails to capture mutual reinforcement between themselves. Doc2vec (Le & Mikolov, 2014) generalizes the idea of vector representation for documents as a stand-alone method. Though it is expensive to infer new document vectors during the test process, and regardless of assigning PV to each word in the document as a single topic, it outperforms BOW (Zhang et al., 2010) and LDA (Fan et al., 2008) in various text understanding tasks. Encouraged by word2vec and Doc2vec, Niu et al. (2015) introduced Topic2Vec. It incorporates the LDA topic model into NPLM (Bengio et al., 2003). However, Topic2Vec cannot capture the polysomic characteristics of terms as LDA models topic occurrence almost independently without capturing topic correlations (He et al., 2017). The purpose of conducting this research is to introduce an efficient multi topical document representation with flowing objectives:

- Firstly, learn probabilistic topical word embeddings using document vector as a global context where each word can be part of different underlying topics of the document instead of just a single paragraph vector. Thus, avail to capture polysemous terms.

- Secondly, embedding documents by simply averaging the topical word embeddings instead of imposing an expensive inference method to learn the document embedding separately.

Unlike clustering-based document modeling with the ability to capture multi-sense of words (Gupta et al., 2019), here the challenge is to learn polysemous words in a jointly learning framework for topic-specific word and document embedding. Capturing polysemous terms is vital because the same word around different context words can represent different themes (Liu et al., 2020) and without noticing them can mislead to an unexpected thematic representation of a document. The Jointly Learning Word Embeddings and Latent Topics (Shi et al., 2017) comes to the rescue for addressing the polysemous word problem, which learns different topical word embeddings using

Skip-gram Topical word Embedding (STE) framework. STE gets the term distributions of the topics and the topic distributions of the document using PLSA(Hofmann, 2013) like mixture weighting for each data point. However, STE doesn't deal with document embedding in its jointly learning process.

In this research, we propose a novel document embedding framework called TDE: Topical Document Embedding. TDE learns topical word and document embeddings jointly. It incorporates the STE-diff framework (Shi et al., 2017) and follows the simple averaging of word embedding with a corruption mechanism for document embedding introduced by Chen (2017).

## 2 MOTIVATION AND RELATED WORK

Huang et al. (2012) proved that combining local and global context performs better for predicting target words in the neural network-based language modeling, but this model doesn't support topic-specific word embeddings. Liu et al. (2015) proposed topic-specific word embeddings where document embeddings get by aggregating over all topical-word embeddings of each word in the document. Arora et al. (2017) extend embedding to the sentence level by smooth inverse frequency (SIF) weighted averaging of the words in a sentence. However, this model is lack of addressing multiple sense of words. Moody (2016) introduced a hybrid model called lda2vec, which suffers from uncorrelated topics due to the incorporation of LDA. For word sense disambiguation with correlated topic modeling, Chaplot & Salakhutdinov (2018) replaced the LDA topic proportions by synset proportions for a document. Other than LDA topic modeling, Arora et al. (2018) have introduced a Gaussian random walk model for word sense disambiguation. This work can address polysemous word representation but doesn't cover document-level embeddings. Mekala et al. (2016) introduced the Gaussian mixture model-based soft clustering method called SCDV, which incorporates both the topic and word embeddings. However, SCDV is not suitable for multisentence documents. Hence Luo et al. (2019) proposed P-SIF, which is fit for large documents with multiple sentences while addressing the underlying topics as well. Though P-SIF overcomes SCDV's shortcomings, it is still a two-step process.

From the motivation of the above advantages and drawbacks, we introduce a novel document modeling capable of learning document embedding simultaneously during the jointly learning process of topic-specific word embeddings. This model creates a pivot by concatenating a document vector and center word vector to predict the context word vector. Here the document vector works as a memory backup for missing information. The output document embeddings produced by this model is more expressive as it represents each document into a higher dimensional embedding space similar to P-SIF.

## 3 MODEL DESCRIPTION

Inspired by the Skip-gram Topical word Embedding (STE) (Shi et al., 2017) and the simplicity of the Document Vector through Corruption(Chen, 2017), we design a novel document embedding model called TDE: Topical Document Embedding. In contrast to Doc2VecC, TDE constructs global context (document vector) as a weighted averaging of the randomly selected topic-specific word embeddings. Unlike STE, TDE uses a generating function, which predicts context words by given the topics and pivot. Pivot constructs by concatenating global context (document vector) and center word (Figure1). The probability of predicting a context word depends on the inner product of the topic-specific embedding of that word (context word) and the pivot, which ensures the participation of underlying word, topic, and document levels knowledge.

### 3.1 TOPICAL DOCUMENT EMBEDDING

A matrix $D$ represents the documents in the corpus, $D = d_1, d_2, \ldots, d_n$ of size $n$. Each document $d$ contains a variable-length sequence of words $W = w_1, w_2, \ldots, w_{T_d}$. Each word $w$ in the vocabulary $V$ of size $v$ of the corpus is generally associated with an input projection matrix $U \epsilon R^{h \times v}$ of size $h$ and an output projection matrix $V^T \epsilon R^{v \times h}$ of size $h$. The input projection matrix represents the input to the hidden space, whereas the output projection matrix represents the hidden to output space. Matrix $V^T$ has the same dimension as matrix $U$ but a transpose matrix. The column $u_w$ of matrix $U$ is the input vector of word $w$, and the column $v_w$ of matrix $V^T$ is the output vector of word $w$.

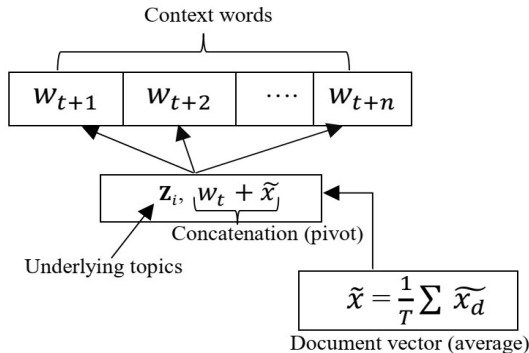

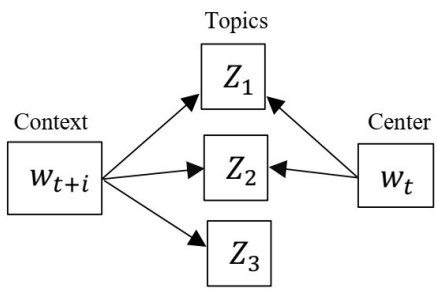

Figure 1: Predicting context words by giving pivots and underlying topics of the document.

Figure 2: Topics assignment of polysemous words.

TDE follows an unbiased mask-out/drop-out corruption mechanism in each update of its stochastic process for randomly select topic-specific word embeddings to create the global context. Under corruption mechanism, each dimension of the original document $d$ overwrites with the probability $q$ and sets the uncorrupted dimensions $\frac{1}{(1-q)}$ times of their original values to make the corruption unbiased. The corruption mechanism reduces word-level noise in the document by forcing the embeddings of common and non-discriminative ones close to zero. This corruption mechanism leads TDE to speed up the training process by significantly pruning the number of parameters for the backpropagation mechanism, document vectors represented by only embeddings of remaining words. The resultant global context is thus as below:

$$\tilde{x}_d = \begin{cases} 0, & with\,probability\,q \\ \\ \frac{d}{1-q}, & otherwise \end{cases} \tag{1}$$

$$Document\,vector\,(global\,context)\,\tilde{x} = \frac{1}{T}\sum \tilde{x}_d \tag{2}$$

where $T$ is the length of the document (number of words) and the document vector $\tilde{x}$ is the weighted average of the words in a document, weighted by the corruption mechanism mentioned above.
The probability of observing the context word $w_{t+i}$ given the center word $w_t$ as well as the global context $\tilde{x}$ and the topics $z_{t+i}$ and $z_t$ for context and center word respectively as

$$p(w_{t+i}\,|\,w_t,\,z_{t+i},\,z_t,\,\tilde{x}) = \frac{exp\left(V_{w_{t+i},\,z_{t+i}}^T \cdot (U_{w_t,\,z_t} + \frac{1}{T}U_{\tilde{x}_d})\right)}{\sum_{w'\,\epsilon\,\Lambda}\,exp\left(V_{w',\,z_{t+i}}^T \cdot (U_{w_t,\,z_t} + \frac{1}{T}U_{\tilde{x}_d})\right)} \tag{3}$$

where $\Lambda$ is the vocabulary of the whole corpus and input projection matrix constructs as $U = U_{w_t,\,z_t} + \frac{1}{T}U_{\tilde{x}_d}$ .
The denominator in equation (3) is generally intractable due to the term $\sum_{w'\,\epsilon\,\Lambda}$. As such, the computational cost is proportional to the size of $\Lambda$. To deal with this intractability requires an efficient approximation method. TDE employs the approximation with the negative sampling (Mikolov et al., 2013a; Chen, 2017). In TDE, the probability of predicting a context word $w_{t+i}$ in document $d$ depends on the pivot $w_t\tilde{x}$ and the topic distributions of the document. The pivot constructed by concatenating the center word $w_t$ and the global context $\tilde{x}$ as-

$$Pivot = w_t\tilde{x} = w_t + \tilde{x} \tag{4}$$

So, for every skip-gram $\langle w_t, w_{t+i}\rangle$, where the topic assignment $z_{t+i}$ of $w_{t+i}$ is independent of the topic assignment $z_t$ of $w_t$ we have :

$$p(w_{t+i}\,|w_t\tilde{x},\,d) = \sum_{z_t=1}^{k}\sum_{z_{t+i}=1}^{k}\,p(w_{t+i}\,|\,w_t\tilde{x},\,z_{t+i},\,z_t)\,p(z_{t+i},\,z_t\,|d)$$

$$= \sum_{z_t=1}^{k}\sum_{z_{t+i}=1}^{k}\,p(w_{t+i}\,|w_t\tilde{x},\,z_{t+i},\,z_t)\,p(z_t|d)p(z_{t+i}\,|d) \tag{5}$$

Following a similar analogy as STE-diff, equation (5) assumes that the context word and the center word may come from the same topic or may come from two different topics as well without imposing any hard constraint (Figure 2). Here, multiple meanings of a word depend on their contexts instead of directly $k$ topics disjoint subspace partitioning as commonly does in clustering-based topic modeling(Gupta et al., 2016). In TDE, word embeddings from different topics co-exist in a common shared space. Instead of the free flow of scattered words in the embedding space, different topical word embeddings stay close together document-wise rather than topic-wise gathering. Creating pivots by concatenating the center word with the document vector provides extra guidance as a global context with a complete thematic sense of the document. Thus, TDE represents more consistent topical word embeddings over STE-diff. In TDE, the compositionality of words encodes in the learned embeddings; as a result, the requirement of heuristic weighting for test documents is no more required; each output document vector is represented simply as an average of the topic-specific word embeddings. Given the learned projection matrix $U_{w,z}$, TDE forms document embedding $d_{vec}$ as below:

$$d_{vec} = \oplus_{z=1}^{k} \frac{1}{T} \sum_{w \, \epsilon \, D} U_{w,z} \tag{6}$$

where $T$ is the length (number of words) of the document and $k$ represents the underlying topics of the document.

In equation (6), the term $\oplus_{z=1}^{k}$ represents topic-wise concatenation. In this case, words in the same topic are just simple average as they represent the same thematic meaning. However, words in different topics carry different themes. Thus, concatenate over topics increases the expressive power of a document. This strategy represents documents into a higher $k \times s$ dimension, where $k$ is the underlying topics, and $s$ is the dimension of the word embeddings.

### 3.1.1 ALGORITHM DESIGN

The inference framework of TDE (Algorithm 1) is similar to STE-diff, based on the EM algorithm with negative sampling (Shi et al., 2017). In each iteration, it updates the word embeddings first, then based on updated word embeddings, updates the document embedding (global context) $\tilde{x}$. Finally, evaluate the topic distribution $p(z \,|\, d)$ of each document. To update word embeddings, iterates over each skip-gram $\langle w_t, w_{t+i} \rangle$ using negative sampling and evaluate the posterior topic distribution. So, the estimation of the probability of predicting the context word $w_{t+i}$ given pivot $w_t \tilde{x}$ and topic assignment $z$ denoted by $p(w_{t+i} \,|\, w_t \tilde{x}, \, z)$ as follows:

$$\log p(w_{t+i}|w_t\tilde{x}, z) \, \propto \, \log \sigma \left( V_{w_{t+i}, \, z}^{T} \cdot (U_{w_t, \, z} \, + \, \frac{1}{T} U_{\tilde{x}_d}) \right)$$
$$+ \sum_{i}^{n} E_{w_i \sim p} \left[ \log \sigma \left( -V_{w_i, \, z}^{T} \cdot (U_{w_t, \, z} + \frac{1}{T} U_{\tilde{x}_d}) \right) \right] \tag{7}$$

where $\sigma(x) = \frac{1}{1+exp(-x)}$ and $w_i$ is a negative instance, which is sampled from the unigram distribution $p(.)$ over terms in the vocabulary and has a value similar to Gupta et al. (2019)'s findings.

The overall training objective is how well the model can predict surrounding words, which is for each document d, given a sequence of words $w_1, w_1, \ldots, w_{T_d}$, then the log-likelihood $\mathcal{L}_d$ is defining as

$$\mathcal{L}_d = \sum_{t=1}^{T_d} \sum_{-c \leq i \leq c; \, i \neq 0} \log p(w_{t+i} \,|\, w_t \tilde{x}, \, d) = 0 \tag{8}$$

where $c$ is the size of the training windows.

The overall log-likelihood $\mathcal{L}$ is then

$$\mathcal{L} = \sum_{d} \mathcal{L}_d \tag{9}$$

The gradients of the objective function to evaluate $\log p(w_{t+i} \,|\, w_t \tilde{x}, \, d)$ for each term concerning the input and the output projection matrix $U$ and $V^T$ respectively can formulate as follows:

$$\frac{\partial \mathcal{L}}{U_{w_t \, \tilde{x}, \, z}} = - \left( \xi_{w' \, w_t} - \sigma(V_{w', z}^{T} \, U_{w_t \tilde{x}, z}) \right) \cdot V_{w', z}^{T} \cdot p(z \,|\, d, \, w_t \tilde{x}, \, w') \tag{10}$$

$$\frac{\partial \mathcal{L}}{V^T_{w',z}} = - \left( \xi_{w'w_t} - \sigma(V^T_{w',z} \, U_{w_t\tilde{x},z}) \right) . \, U_{w_t \, \tilde{x},z}. \, p(z \,|\, d, \, w_t\tilde{x}, \, w') \tag{11}$$

where

$$\xi_{w'w_t} = \begin{cases} 1, & if \; w' \, in \, the \, context \, window \, of \; w_t \\ 0, & otherwise \end{cases} \tag{12}$$

In TDE, the update rule of $U$ and $V^T$ maximizes $p(w_{t+i}|w_t\tilde{x}, z)$, thus $\log p(w_{t+i}|w_t\tilde{x}, z)$ for each skip-gram $\langle w_t, w_{t+i} \rangle$ instead of maximizing $\log p(w_{t+i} \,|\, w_t, z)$ as STE does. This observation shows both of the models flow the same procedure. STE has proved capable of discovering good topics and identifying high-quality word embedding vectors. Thus, TDE inherits all the prominences; moreover, it keeps Doc2VecC's simplicity for document embeddings and extends reliability by filling up essential information gaps using global document context. When concerning topics for the embedding process, the generally used hierarchical Softmax or Negative sampling methods are incapable of obtaining the embeddings efficiently. The EM-negative sampling Algorithm thus comes to the rescue.

In the E-step, the posterior topic distribution for each skip-gram $\langle w_t, w_{t+i} \rangle$ in document $d$ evaluate using the Bayes rule:

$$p(z'_k \,|\, d, w_t\tilde{x}, \, w_{t+i}) = \frac{p(w_{t+i} \,|z'_k, w_t\tilde{x})p(z'_k \,|\, d)}{\sum_z p(w_{t+i} \,|z, w_i\tilde{x})p(z \,|\, d)} \tag{13}$$

In the M-step, given the posterior topic distribution, it is now time to update the topic-specific word embeddings and the topic distribution of the document by maximizing the following $Q$ function:

$$\begin{aligned} Q &= \sum_d \sum_{t=1}^{T_d} \sum_{-c \le i \le c; \; i \ne 0} \sum_z p(z \,|\, d, w_t\tilde{x}, \, w_{t+i}) \log(p(z \,|\, d)p(w_{t+i} \,|\, z, \, w_t\tilde{x})) \\ &= \sum_d \sum_{\{w_t, \, w_{t+i}\} \, \epsilon \, P_d} n(d, \, w_t, \, w_{t+i}) \sum_z p(z \,|\, d, w_t\tilde{x}, \, w_{t+i}) \left[ \log(z \,|\, d) + \log(p(w_{t+i} \,|\, z, \, w_t\tilde{x})) \right] \end{aligned} \tag{14}$$

Where $P_d$ is the set of skip-gram in document $d$ and $n(d, \, w_t, \, w_{t+i})$ denotes the number of skip-gram $\langle w_t, w_{t+i} \rangle$ in document $d$.

For each document $d$, the prior topic distribution $p(z \,|\, d)$ can be updated using the Lagrange multiplier while keeping the normalization constraints i.e. $\sum_z p(z \,|\, d) = 1$.

$$p(z \,|\, d) = \frac{\sum_{\{w_t, w_{t+i}\} \, \epsilon \, P_d} n(d, \, w_t, \, w_{t+i})p(z \,|\, d, \, w_t\tilde{x}, \, w_{t+i})}{\sum_{\{w_t, w_{t+i}\} \, \epsilon \, P_d} n(d, \, w_t, \, w_{t+i})} \tag{15}$$

The EM negative sampling algorithm in TDE follows a similar workflow as in STE. TDE can evaluate the likelihood estimation $p(w_{t+i} \,|\, z, \, w_t\tilde{x})$ for each skip-gram $\langle w_t, w_{t+i} \rangle$. As a result, each topic represents by the ranked list of the bi-grams, similar to STE. As the document vector is learning simultaneously with word embeddings by averaging the word vectors, the time complexity of $p(w_{t+i} \,|\, z, \, w_t\tilde{x})$ is thus $|\Lambda|^{2 \times K}$, where $|\Lambda|$ is the size of vocabulary in the document and $K$ is the number of topics, which is the same as STE.

For each word $w$ in a given new document $d'$, the posterior topic distribution infers by considering the topic distribution of the context words $w_c$ and the document $d'$, while keeping the values of the learned input and output projection matrices $U$ and $V^T$ respectively unchanged. As instead of the center word $w$ itself, we use pivot $w\tilde{x}$. Therefore, using the Bayes rule:

$$p(z \,|\, w\tilde{x}, \, w_c, \, d') \propto \log(z \,|\, d') + \log p(w_c \,|\, z, \, w\tilde{x}, \, d') \tag{16}$$

where $w_c$ is the set of the context words of word $w$, and $w\tilde{x}$ is the pivot created by concatenating global context $\tilde{x}$ with the center word vector $w$. The log-likelihood term $\log p(w_c \,|\, z, \, w\tilde{x}, \, d')$ can define as the sum of $\log p(w_c \,|\, w\tilde{x}, z)$, and $\log(z|d')$ is the corresponding prior probability. The probability $\log p(w_c \,|\, w\tilde{x}, \, z)$ can be computed using Eq.7

The output document vector creates directly using Eq.6 without imposing an inference method for the document separately.

---

**Algorithm 1** TDE using EM negative sampling

---

1: Initialize Input and Output projection matrix $U$ and $V^T$ respectively and the topic distribution of the document $p(z \mid d)$.
2: **for** $out\_iter = 1$ to $Max\_Out\_iter$ **do**
3:   **for** each document $d$ in $D$ **do**
4:     **for** each word $w$ in $U$ **do**
5:       Create document vector (global context) $\tilde{x}$ using Eq.1 and Eq.2
6:     **end for**
7:     **for** each skip-gram $\langle w_t, w_{t+i} \rangle$ in $d$ **do**
8:       Sample negative instances from the distribution P.
9:       Construct pivot $w_t\tilde{x}$ using Eq.4
10:       Update likelihood function $p(w_{t+i} \mid w_t\tilde{x},\, z)$ using Eq.7 and posterior topic distribution $p(z'_k \mid d,\, w_t\tilde{x},\, w_{t+i})$ using Eq.13
11:       **for** $in\_iter = 1$ to $Max\_in\_iter$ **do**
12:         Update projection matrix $U$ and $V^T$ using gradient decent method with Eq.10 and Eq.11 respectively.
13:       **end for**
14:     **end for**
15:     Update topic distribution $p(z \mid d)$ using Eq.15
16:   **end for**
17: **end for**
    /* Creating document vectors using test dataset */
18: **for** each test_document $d'$ in $D'$ **do**
19:   **for** each word $w$ in updated $U_{w,z}$ **do**
20:     Construct the output document vector $d_{vec}$ using Eq.6
21:   **end for**
22: **end for**

---

Table 1: Document classification using linear SVM (libliner-2.30) classifier

| Models | Number of Train docs | Number of Test docs | Accuracy (%) | Error rate (%) |
|---|---|---|---|---|
| TDE | 50,000 | 50,000 | **75.00** | **25.00** |
| | 100,000 (all) | 100,000 (all) | **87.50** | **12.50** |
| Doc2VecC | 50,000 | 50,000 | 72.29 | 27.71 |
| | 100,000 (all) | 100,000 (all) | 87.32 | 12.68 |

## 4 EXPERIMENTS AND ANALYSIS

The purpose of this research is to implement a simple but effective document embedding method. This section is dedicated to demonstrate a detailed analysis of the experiments and obtained results. To analyze document embeddings obtained by the proposed model (TDE), we use a linear SVM classifier to check the performance of the document classification problem. We also extend our analysis to topical word embeddings efficiency check as we proposed to utilize document vector as a global context in the process of jointly learning word-topic embeddings.

### 4.1 DOCUMENT CLASSIFICATION FOR DOCUMENT EMBEDDING EVALUATION

Doc2VecC doesn't consider topic-specific words in its document embeddings; it is thus in the same embedding space as the word. On the other hand, TDE represents documents into higher ($number\ of\ clusters \times number\ of\ word\ embeddings\ features$) dimensional space.
We use a subset of the Wikipedia dump (Chen, 2017), which contains over 300,000 Wikipedia pages in 100 categories. In our experiments both TDE and Doc2VecC learn 100,000 unique documents representation. For both models, we set embedding size to 400, window size to 10, negative sampling to 8, subsampling frequent words to 10e-4, number of threads to 10, sentence sample to 0.1, and choose liblinear-2.30 as a linear SVM classifier (Fan et al., 2008), for TDE we set number of topics to 10, inner and outer iteration to 15. Table 1 shows both TDE and Doc2VecC (base model) perform better classification on seen data than unseen data. For TDE and Doc2VecC, we gave the

first 50,000-document vectors as a train set; and the last 50,000 document vectors as a test set, where TDE gives 75% accuracy with a 25% error rate and Doc2VecC exhibits 72.29% accuracy with a 27.71% error rate. When we provide all 100,000 document vectors for both train and test, TDE gives 87.5% accuracy with a 12.5% error rate, and Doc2VecC shows 87.32% accuracy with a 12.68% error rate. In Table 1, highlighted cells represent better scores obtained by TDE.

## 4.2 WORD EMBEDDING EVALUATION

For word embedding evaluation, we use Stanford Contextual Word Similarity (SCWS) data set (Huang et al., 2012). SCWS dataset contains 2003 word pairs; most of them are monosemous words, only 241-word pairs containing the same term within a word pair among them only 50 have a similarity score less than 5.0. That indicates a small number of challenging polysemous words in the SCWS dataset. To obtain word embeddings for both STE-diff and TDE, we use the 20Newsgrup dataset to train the models. For both models, we set embedding size to 400, number of topics to 20, window size to 10, negative sampling to 8, subsampling frequent words to 10e-4, number of threads to 10, both inner and outer iteration to 15, and minimum word count to 5. We take 1693 word pairs under consideration from the SCWS dataset (commonly found in the 20Newsgrup and SCWS datasets) for Spearman's rank correlation coefficient measures between models scores (achieved by TDE and STE-diff) and the human judgments of similarity scores in the SCWS dataset. For each pair of words, pairwise cosine similarity uses for the word similarity evaluation of the base (STE-diff) and the proposed(TDE) model; for both models, we use two variants of similarity scoring-AgvSimC and MaxSimC (Shi et al., 2017). Table 2 exhibits obtained scores by the base and the

Table 2: Spearman correlation ($\rho \times 100$)

| Model | Similarity Metrics | $\rho \times 100$ |
|-------|-------------------|-------------------|
| TDE | MaxSimC | **39.22** |
|  | AvgSimC | **39.04** |
| STE-diff | MaxSimC | 31.56 |
|  | AvgSimC | 33.28 |

Table 3: PMI scores as topic coherence

| Model | T=5 | T=10 | T=15 | T=20 |
|-------|-----|------|------|------|
| TDE | **-69.36** | **-331.74** | **-729.65** | **-1349.46** |
| STE-diff | -85.23 | -374.84 | -875.77 | -1599.42 |

proposed model under the conditions mentioned above. According to Table 2, TDE obtains almost similar scores for MaxSimC (39.22) and the AvgSimC (39.04) whereas, in STE-diff, MaxSimC gets 31.56 and AvgSimC gets 33.28. Highlighted cells indicate better scores in the table obtained by TDE.

## 4.3 WORD EMBEDDING VISUALIZATION

To visualize the interaction between words and latent topics in the embedding space, we use the Wikipedia dump from April 2010 (Shaoul, 2010), similar to STE. For both base (STE-diff) and proposed (TDE) models, we set embedding size to 400, number of topics to 10, window size to 10, negative sampling to 8, subsampling frequent words to 10e-4, number of threads to 10, the inner and the outer iterations to 15. For visualization in Fig 3 and Fig 4, we use the t-SNE algorithm (Van der Maaten & Hinton, 2008), where each node represents a topic-specific word vector. For both models, we set perplexity to 40, n_iteration to 2500, and random state to 23. We consider the word *party* with two different meanings *political party* and *social gathering* as a polysemous word for visualization. In Fig 3, party#9 is close to government#9 indicating political party but party#7 is far from government#9. our extensive analysis found party#0, party#1, party#2, party#6, and party#9 contain government#9 and government#0 within topmost 100 similar words, whereas party#3, party#4, party#5, party#7, party#8 are close to word summers#5, indicating a gathering and celebration. These words contain summer#5 in the topmost 50 similar words.
In Fig 4, It seems party and government exist little scattered in the embedding space for STE-diff than TDE; it indicates learning better embeddings by TDE over STE-diff. Our extensive experiment found, only party#6 contains government#6 within the topmost 100 similar words, whereas summer#4 exists in the most 150 similar words of party#4 only.

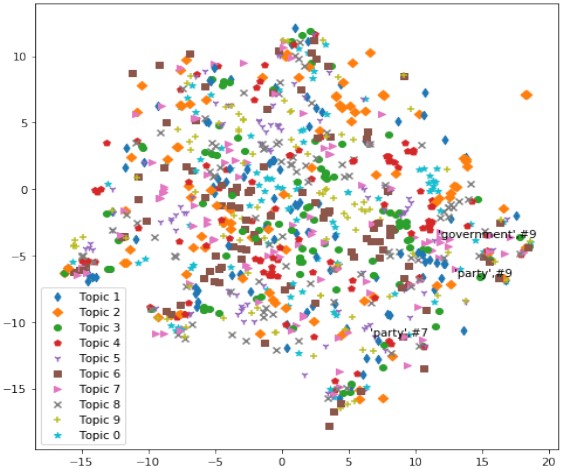

Figure 3: TDE polysemous word embedding visualization.

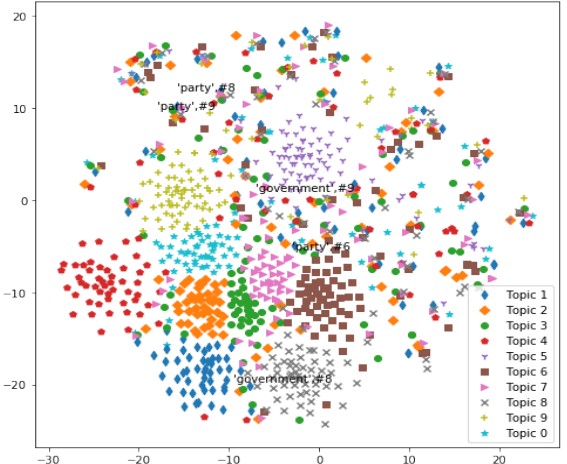

Figure 4: STE-diff polysemous word embedding visualization.

Table 4: TDE topics generating bigrams for 20Newsgrups dataset

| god heaven | image files | mr president |
|---|---|---|
| jesus died | open file | national security |
| believe god | file format | national institute |
| jesus god | file server | president clinton |
| jesus christ | file manager | clinton administration |
| father son | image processing | senior administration |
| father jehovah | file size | national center |
| god father | file stream | press conference |
| jehovah elohim | output file | bill clinton |
| christ god | postscript file | press release |
| alt.atheism | comp.graphics | talk.politics.misc |
| soc.religion.christian | comp.os.ms-windows.misc | talk.politics.guns |
| talk.religion.misc | comp.sys.ibm.pc.hardware | talk.politics.midest |
| | comp.sys.mac.hardware | |
| | comp.windows.x | |

## 4.4 TOPIC COHERENCE

In TDE, pointwise mutual information (PMI) scores are using to evaluate topic coherence as PMI scores strongly correlate with human annotations of topic coherence (Newman et al., 2010; Lau et al., 2014). For this experiment, we use the 20Newsgrup dataset. The parameter setting for both models base (STE-diff) and proposed (TDE) are the same as mentioned in section 4.2. Table 3 exhibits overall PMI scores for different numbers of top words (T) under 20 topics. The highlighted cells in the table indicate higher coherence, as we obtained negative coherence scores so, the smaller is better.

Topics in TDE can be interpreted by bigrams words as shown in the publication (Shi et al., 2017) for STE-same. In our experiments, we found for STE-diff, each topic represents bigrams randomly

Table 5: Multi-class document classification for word embedding evaluation.

| Model | Accuracy | Precision | Recall | F-measure |
|-------|----------|-----------|--------|-----------|
| PV | 75.4 | 74.9 | 74.3 | 74.3 |
| STE-same | 80.4 | 80.3 | 80.4 | 80.2 |
| TDE | 80.5 | 80.5 | 80.5 | 80.3 |
| STE-diff | 82.9 | 82.5 | 82.3 | 82.5 |

from various categories without thematic grouping of bigrams, whereas TDE maintains thematic gathering of bigrams. Table 4 represents three topical representations of bigram words with their source categories for TDE. The first column represents a religion-related topic, the second is related to the computer-related topic, and the third is related to politics.

## 4.5 DOCUMENT CLASSIFICATION FOR WORD EMBEDDING EVALUATION

In this experiment, we create document vectors by TF-IDF weighted averaging of each word embeddings under every document, using equation (17). For both models, the base (STE-diff) and the proposed (TDE), we use the 20Newsgroup dataset with the same parameter settings mentioned in section 4.2.

$$d_w = tfidf_w \times \sum_{z=1}^{k} p(z \mid w\tilde{x}, c).U_{w,z} \tag{17}$$

where $U_{w,z}$ is the learned input projection matrix, $c$ represents context words, $w\tilde{x}$ represents pivots (concatenating center words $w$ with the global document context $\tilde{x}$), and $tfidf_w$ is the TF-IDF score of $w$ in the document.

For document classification in this section, we use several standard evaluation metrics for the multi-class document classification tasks, e.g., Accuracy, Precision, Recall, and F-measure for LinearSVM classifier. We compare scores obtained by TDE with the obtained scores by PV, STE-diff, and STE-same models published by Shi et al. (2017). Table 5 represents obtained scores using evaluation metrics in the classification task, where TDE beats STE-same and PV, but STE-diff performs better in this case. TDE achieves better topic coherence than STE-diff, but STE-diff achieves better document classification, which we see a similar analogy between STE-same and STE-diff in Shi et al. (2017)'s work.

## 5 CONCLUSION

TDE: Topical Document Embedding is a jointly learning word-topic and document embeddings method, which learns document embedding on the fly simultaneously with the jointly learning process of the topic-specific word embeddings by taking document vector as a global context. As a result, TDE captures better syntactic and semantic properties by utilizing word, topic, and document-level information. It poses the simplest form of document embedding without imposing complex inference methods for document vectors. As such, each document vector is created by a simple weighted averaging of the topic-specific word embeddings, which is effective regardless of document size. Moreover, using the mask-out/drop-out corruption mechanism enhances learning speed and ensures quality embeddings as well. Our extensive experimental findings reveal that besides the document embeddings, the proposed joint learning method also captures better topical word embeddings over the base model.

## 6 REPRODUCIBILITY STATEMENT

Model source code written in C programming language, to run the code need to type *./go.sh* in the terminal (optional: for creating make file type *make CFLAG="-lm -pthread -O3 -Wall -funroll-loops"*) . Model parameters can be reset in this file (e.g., training dataset). All datasets used in this research are available in preprocessed form (use ready for the model) in the *data* folder of the package (Model_sourceCode_Datasets). The model output should be reproduced effortlessly in a Linux machine with *GCC C++* compiler installed.

For model evaluaiton, we have added evaluation source code files *ste_d2vc_scws_similarity.py* and *visualize.ipynb*. The evaluation procedure requires python-2.7, numpy-1.16.1, scikit-learn-0.17. For convenience, we also included obtained evaluation results and detail analysis for embedding visualization and pearsoń correlation in the *obtained_results* subfolder for both base and proposed models.

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
