# OpenReview forum: "JOINTLY LEARNING TOPIC SPECIFIC WORD AND DOCUMENT EMBEDDING"
_ICLR.cc/2022/Conference — ICLR 2022 Submitted_

### Official Review · Reviewer_iEk8 · 2021-10-26

**Correctness:** 2
**Technical Novelty And Significance:** 2
**Empirical Novelty And Significance:** 2
**Recommendation:** 3
**Confidence:** 4

**Main Review:**

- The proposed TDE method is just a simple mix of the ideas of Doc2VecC proposed by (Chen 2017) and STE proposed by (Shi et al., 2017).

- Although the paper clearly explains the motivation and the related work section has a good collection of references on conventional word embeddings and document embeddings, most references (including two baselines) are publications of 4 or 5 years ago and completely misses recent major progresses of contextual sentence/document embeddings since ELMO and BERT.

- What's the advantage of the proposed TDE over the recent contextual embedding methods for representation of sentences and documents?

-  Both Doc2VecC (Chen 2017) and STE (Shi et al., 2017) conducted extensive experiments on the quality of the embeddings and performance on the downstream application of document classification. But this paper only presents very limited experiments and is lack of clear interpretation of experimental results. For example, there are only 2 base lines (STE and Doc2VecC) studied in the experiments, but why STE is still missing Table 1, and Doc2VecC is missing in Table 5? What's the key observations and insights from Table 4?

- It seems there is lack of basic check and proof reading
    - Typos: for example, "Tow popular complementary" in Page 1 and "flowing objectives" in Page 1
    - error on major equation: for example: I guess “d/1-q” should be "x_d/1-q" in equation (1)
    - Missing definition: for example, "PV" in Page 1 and other pages (I guess "PV" means "paragraph vector")
    - Poor Notations and writings, hard to read and follow, for example, section 3.1


**Summary Of The Paper:**

- This paper presents a document-embedding method called Topical Document Embedding (TDE) which utilizes Document Vector through Corruption (Doc2VecC) proposed by (Chen 2017) as the basic framework and combines the idea of Skip-gram Topical word Embedding (STE) proposed by (Shi et al., 2017) for further enhancement.

- The authors claimed that the proposed TDE demonstrates better topical word embeddings using document vector as a global context and better document classification results on the obtained document embeddings.


**Summary Of The Review:**

In general, this paper has not been carefully checked and well organized, and the experimental evaluations are not solid. This paper needs some major updates for future improvements.

---

### Official Review · Reviewer_Nesz · 2021-10-28

**Correctness:** 2
**Technical Novelty And Significance:** 1
**Empirical Novelty And Significance:** 1
**Recommendation:** 3
**Confidence:** 5

**Main Review:**

strengths
1) This paper is well written and easy to read.
2) TDE follows the previous work and aims to learn three different level embeddings.
3) This paper shows visualization of the interaction between words and latent topics in the embedding space.

weaknesses
1) Approaches are straightforward and lack originality. This approach is simple combination of  Doc2Vec and STE.
2) Baselines are old and weak in now.
3) The experimental results reported are validated on a single dataset,
and no human evaluation and error analysis.

**Summary Of The Paper:**

This paper proposes Topical Document Embedding, TDE,  to obtain document vectors on during the jointly learning process of the topical word embeddings,
where it captures syntactic and semantic properties by utilizing three levels of knowledge (i.e., word, topic, and document).
Experiments demonstrate better topical word embeddings using document vector and better document classification results on the obtained document embeddings by the proposed method over the recent related models.

**Summary Of The Review:**

Now, the mainstream of representation learning is attention based models or Transformer based models,
so we would like to see a comparison with these and discussion.
For example, we use BERT and then get a similar expression from [CLS] token.

---

### Official Review · Reviewer_PLvo · 2021-11-01

**Correctness:** 3
**Technical Novelty And Significance:** 1
**Empirical Novelty And Significance:** 1
**Recommendation:** 3
**Confidence:** 4

**Main Review:**

The proposed Skip-gram model contains two major parts that are different from the standard Skip-Gram model:

* One is to assume each word has a different representation under each individual topic. In such a way, the topic assignment of a word can be factored in the computation of the skip-gram probability. In other words, the probability of a context word will depend on not only the centre word but also their topic assignments. However, this idea is from the Skip-gram topical embeddings by Shi et al, 2017.

* The other is the use of a document vector as the global context, which originates from Chen 2017. The proposed model simply injects this context vector into the Skip-gram topical word embedding.  If one compares eq (4) in Shi et al, 2017 and eq (3) of this paper, the only difference is the term $\frac{1}{T} U_{\tilde{x}_d}$, constructed similarly to Eq (2) of Chen 2017. This simple modification makes the learning algorithm naturally follow the algorithm design discussed in Section 3.2 of Shi et al, 2017.

There is nothing wrong with combining the idea of two models together. However, it makes one have doubt about the innovation of this work unless the proposed model is very effective in terms of performance, which is not really the case.

In regards to the experiments, the proposed model is compared mainly compared with the models in Shi et al, 2017. The experimental methodology follows closely that used by Shi et al, 2017, i.e. document classification, topic coherence, word embedding. The classification performance is only marginally better than STE-same, much worse than STE-diff. Meanwhile, the paper claims that the proposed model is capable of dealing with polysemy, which however is only insufficiently demonstrated by Figures 3 and 4.

The writing of this paper could be further improved, particularly those notations.

Besides, I have the following questions
* Should the two project matrices be in $\mathcal{R}^{k \times s}$?  If that is the case, most likely, then it column is not a word embedding. I understand if you follow the notation of Chen 2017, then the column should be word embeddings, However, the equations tell us that U has a topic dimension. Let me know if I was wrong.
* By Eq (4), do the authors mean that the skip-gram probability conditions on both $w_t$ and $\tilde{x}$? And presumably, the first probability of the left term in Eq (5) is computed based on Eq (3).
* Eq (1): $\tilde{x}$ is scalar. The sum in Eq (2) will be scalar, won't it? According to Chen 2017,  Eq(1) will generate a word masking vector for  a document. The masking vector is then used to generate the document representation via averaging. Right?




**Summary Of The Paper:**

Jointly learning word embeddings and document embeddings while taking into account topical information is somewhat interesting. The paper presents an improved skip-gram model where each individual word is associated with two matrices (e.g., the input projection matrix and the output projection matrix) which is designed to capture the topical aspect of words. In other words, each word is associated with K topic embeddings. While computing the predictive probability, the proposed model further considers a global context, i.e, a document embedding computed with the idea of document corruption. Basically, this paper simply combines the work of Shi et al. 2017 and Chen 2017, Both the innovation and experimental improvement are mariginal.

**Summary Of The Review:**

As discussed above, the proposed method simply inject the idea of the global context in Chen 2017 into the Skip-gram topical embeddings by Shi et al, 2017, which is very incremental. Even regardless of that, the performance gain of the proposed model is very marginal, even worse than STM-diff.

---

### Official Review · Reviewer_a6Hd · 2021-11-03

**Correctness:** 4
**Technical Novelty And Significance:** 2
**Empirical Novelty And Significance:** 2
**Recommendation:** 5
**Confidence:** 4

**Main Review:**

Strengths:
+ introduced a single framework that jointly learns topically-aware word embeddings and document vectors where topic-embeddings are learning using global content and each word embedding can be part of different underlying topics of the document. On top, document vectors are computed by averaging topic word embeddings.
+ clear problem statement
+ paper is well written
+ sound qualitative evaluation of word embedding visualization

Weaknesses:
- missing empirical comparison to related works in compositional models [1, 2, 3] of jointly learning topic and word embeddings
- experimental setup is unclear for document classification task: How is classification task performed? First LM-based objective then using the document representation into classification?
- related works section is weak, missing several related works [1, 2, 3]
- quantitative evaluation is limited (missing key baselines and requires additional data sets in comparative analysis)
- Document classification is performed on a single (large) data set.  Would be interesting to validate the methodology using at least 2-3 datasets and in sparse data settings.
- Document classification should be further evaluated on standard data sets for example 20newsgroups, Reuters, TMN, etc.
- Empirical evaluation is incomplete for document and word embedding evaluation: missing comparison with several strong and most related baselines such as [1, 2, 3] etc. that jointly learns topic and word embeddings
- please include a case study and/or comparison/differences to related works [1, 2, 3] in terms of:
	- jointly learning topic and word embeddings in a single framework
	- captures both the syntactic and semantic linguistic structure
	- combines both local and global contextual information from word, topic and document
	- incorporates both: topic and word embedding spaces in language modeling based training objective (loss function)

Questions:
1. Since the Document vector is computed by averaging its word vectors, therefor the averaging scheme may not consider the syntactic structure e.g. word ordering. However, the methods such as [1, 2] employs LSTM to compute document vectors and globsl context considering linguistic structures.
2. In equation 3, does the global context \~{x} include w_{i+1} ? Unclear.
3. In qualitative evaluation of topics, why are the topics of bigram words only? Please also include qualitative topics analysis of unigram words?

Missing references:
1. Gupta et al., 2019. textTOvec: DEEP CONTEXTUALIZED NEURAL AUTOREGRESSIVE TOPIC MODELS OF LANGUAGE WITH DISTRIBUTED COMPOSITIONAL PRIOR. ICLR 2019.
2. Lau et al., 2017. Topically driven neural language model.  ACL 2017.
3. Dieng et al., 2016. TopicRNN: A recurrent neural network with long-range semantic dependency.

**Summary Of The Paper:**

This paper focuses on learning document embeddings, presenting a topic-document embedding (TDE) method employing syntactic and semantic properties by jointly learning topic and word embedding in a single framework. The proposed TDE approach follows corruption mechanism to create the global context and randomly select topic-specific word embeddings in learning document vectors, thus employing word, topic and document information in learning document vectors.

The experimental results have shown improved performed in terms of document classification, topic coherence and word embedding similarity evaluations.

**Summary Of The Review:**

- quantitative evaluation is weak
- missing related works and quantitative comparison
- incremental work

---

### Decision · Program_Chairs · 2022-01-20

**Decision:**

Reject

**Comment:**

Strength
* The paper is relatively clearly written.
* A new method is proposed.

Weakness
* The evaluation is weak.  The experimental setup is not clear enough.  More quantitative evaluation is necessary. There are strong and new baselines that need to be compared with.
* Relation with existing work needs to be more clearly described.
* The novelty of the work is limited.  It is a combination of existing methods.
* Justification of the proposed method needs to be provided.
* The writing of the paper can be improved.